# Vocos: Closing the gap between time-domain and Fourier-based neural vocoders for high-quality audio synthesis

**Hubert Siuzdak**[*]

## Abstract

Recent advancements in neural vocoding are predominantly driven by Generative Adversarial Networks (GANs) operating in the time-domain. While effective, this approach neglects the inductive bias offered by time-frequency representations, resulting in reduntant and computionally-intensive upsampling operations. Fourier-based time-frequency representation is an appealing alternative, aligning more accurately with human auditory perception, and benefitting from well-established fast algorithms for its computation. Nevertheless, direct reconstruction of complex-valued spectrograms has been historically problematic, primarily due to phase recovery issues. This study seeks to close this gap by presenting Vocos, a new model that directly generates Fourier spectral coefficients. Vocos not only matches the state-of-the-art in audio quality, as demonstrated in our evaluations, but it also substantially improves computational efficiency, achieving an order of magnitude increase in speed compared to prevailing time-domain neural vocoding approaches. The source code and model weights have been open-sourced at `https://github.com/gemelo-ai/vocos`.

## 1 Introduction

Sound synthesis, the process of generating audio signals through electronic and computational means, has a long and rich history of innovation . Within the scope of text-to-speech (TTS), concatenative synthesis (Moulines & Charpentier, 1990; Hunt & Black, 1996) and statistical parametric synthesis (Yoshimura et al., 1999) were the prevailing approaches. The latter strategy relied on a source-filter theory of speech production, where the speech signal was seen as being produced by a source (the vocal cords) and then shaped by a filter (the vocal tract). In this framework, various parameters such as pitch, vocal tract shape, and voicing were estimated and then used to control a *vocoder* (Dudley, 1939) which would reconstruct the final audio signal. While vocoders evolved significantly (Kawahara et al., 1999; Morise et al., 2016), they tended to oversimplify speech production, generating a distinctive "buzzy" sound and thus compromising the naturalness of the speech.

A significant breakthrough in speech synthesis was achieved with the introduction of WaveNet (Oord et al., 2016), a deep generative model for raw audio waveforms. WaveNet proposed a novel approach to handle audio signals by modeling them autoregressively in the time-domain, using dilated convolutions to broaden receptive fields and consequently capture long-range temporal dependencies. In contrast to the traditional parametric vocoders which incorporate prior knowledge about audio signals, WaveNet solely depends on end-to-end learning.

Since the advent of WaveNet, modeling distribution of audio samples in the time-domain has become the most popular approach in the field of audio synthesis. The primary methods have fallen into two major categories: autoregressive models and non-autoregressive models. Autoregressive models, like WaveNet, generate audio samples sequentially, conditioning each new sample on all previously generated ones (Mehri et al., 2016; Kalchbrenner et al., 2018; Valin & Skoglund, 2019). On the other hand, nonautoregressive models generate all samples independently, parallelizing the process and making it more computationally efficient (Oord et al., 2018; Prenger et al., 2019; Donahue et al., 2018).

---

[*]Work conducted while at Gemelo AI

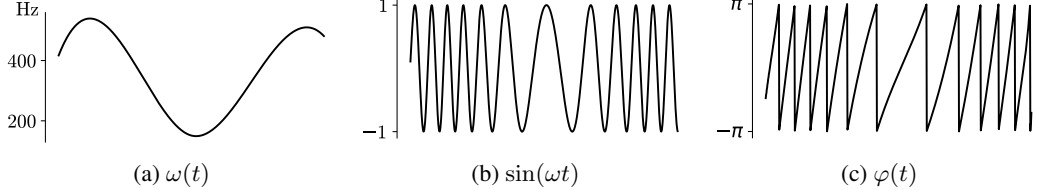

(a) $\omega(t)$    (b) $\sin(\omega t)$    (c) $\varphi(t)$

Figure 1: This illustrates the phase wrapping using an example sinusoidal signal (b) generated with a time-varying frequency (a). The instantaneous phase, $\varphi(t)$, is shown in (c). The apparent discontinuities observed around $-\pi$ and $\pi$ are the result of phase wrapping. Nevertheless, when viewed on the complex plane, these discontinuities represent continuous rotations. The instantaneous phase is computed as $\varphi(t) = \arg\{\hat{s}(t)\}$, where $\hat{s}(t)$ denotes the Hilbert transform of $s(t) = \sin(\omega t)$.

## 1.1 Challenges of modeling phase spectrum

Despite considerable advancements in time-domain audio synthesis, efforts to generate spectral representations of signals have been relatively limited. While it's possible to perfectly reconstruct the original signal from its Short-Time Fourier Transform (STFT), in many applications, only the magnitude of the STFT is utilized, leading to inherent information loss. The magnitude of the STFT provides a clear understanding of the signal by indicating the amplitude of different frequency components throughout its duration. In contrast, phase information is less intuitive and its manipulation can often yield unpredictable results.

Modeling the phase distribution presents challenges due to its intricate nature in the time-frequency domain. Phase spectrum exhibits a periodic structure causing wrapping around the principal values within the range of $(-\pi, \pi]$ (Figure 1). Furthermore, the literature does not provide a definitive answer regarding the perceptual importance of phase-related information in speech (Wang & Lim, 1982; Paliwal et al., 2011). However, improved phase spectrum estimates have been found to minimize perceptual impairments (Saratxaga et al., 2012). Researchers have explored the use of deep learning for directly modeling the phase spectrum, but this remains a challenging area (Williamson et al., 2015).

## 1.2 Contribution

Attempts to model Fourier-related coefficients with generative models have not achieved the same level of success as has been seen with modeling audio in the time-domain. This study focuses on bridging that gap with the following contributions:

- We propose Vocos – a GAN-based vocoder, trained to produce complex STFT coefficients of an audio clip. Unlike conventional neural vocoder architectures that rely on transposed convolutions for upsampling, this work proposes maintaining the same feature temporal resolution across all layers. The upsampling to waveform is realized through the Inverse Fast Fourier Transform.

- To estimate phase angles, we propose a simple activation function defined in terms of a unit circle. This approach naturally incorporates implicit phase wrapping, ensuring meaningful values across all phase angles.

- As Vocos maintains a low temporal resolution throughout the network, we revisited the need to use dilated convolutions, typical to time-domain vocoders. Our results indicate that integrating ConvNeXt (Liu et al., 2022) blocks contributes to better performance.

- Our extensive evaluation shows that Vocos matches the state-of-the-art in audio quality while demonstrating over an order of magnitude increase in speed compared to time-domain counterparts. The source code and model weights have been made open-source, enabling further exploration and potential advancements in the field of neural vocoding.

## 2 RELATED WORK

**GAN-based vocoders** Generative Adversarial Networks (GANs) (Goodfellow et al., 2014), have achieved significant success in image generation, sparking interest from audio researchers due to their ability for fast and parallel waveform generation (Donahue et al., 2018; Engel et al., 2018). Progress was made with the introduction of advanced critics, such as the multi-scale discriminator (MSD) (Kumar et al., 2019) and the multi-period discriminator (MPD) (Kong et al., 2020). These works also adopted a feature-matching loss to minimize the distance between the discriminator feature maps of real and synthetic audio. To discriminate between real and generated samples, also multi-resolution spectrograms (MRD) were employed (Jang et al., 2021).

At this point the standard practice involves using a stack of dilated convolutions to increase the receptive field, and transposed convolutions to sequentially upsample the feature sequence to the waveform. However, this design is known to be susceptible to aliasing artifacts, and there are works suggesting more specialized modules for both the discriminator (Bak et al., 2022) and generator (Lee et al., 2022). The historical jump in quality is largely attributed to discriminators that are able to capture implicit structures by examining input audio signal at various periods or scales. It has been argued (You et al., 2021) that the architectural details of the generators do not significantly affect the vocoded outcome, given a well-established multi-resolution discriminating framework. Contrary to these methods, Vocos presents a carefully designed, frequency-aware generator that models the distribution of Fourier spectral coefficients, rather than modeling waveforms in the time domain.

**Phase and magnitude estimation** Historically, the phase estimation problem has been at the core of audio signal reconstruction. Traditional methods usually rely on the Griffin-Lim algorithm (Griffin & Lim, 1984), which iteratively estimate the phase by enforcing spectrogram consistency. However, the Griffin-Lim method introduces unnatural artifacts into synthesized speech. Several methods have been proposed for reconstructing phase using deep neural networks, including likelihood-based approaches (Takamichi et al., 2018) and GANs (Oyamada et al., 2018). Another line of work suggests perceptual phase quantization (Kim, 2003), which has proven promising in deep learning by treating the phase estimation problem as a classification problem (Takahashi et al., 2018).

Despite their effectiveness, these models assume the availability of a full-scale magnitude spectrogram, while modern audio synthesis pipelines often employ more compact representations, such as mel-spectrograms (Shen et al., 2018). Furthermore, recent research is focusing on leveraging latent features extracted by pretrained deep learning models (Polyak et al., 2021; Siuzdak et al., 2022).

Closer to this paper are studies that estimate both the magnitude and phase spectrum. This can be done either implicitly, by predicting the real and imaginary parts of the STFT, or explicitly, by parameterizing the model to generate the phase and magnitude components. In the former category, Gritsenko et al. (2020) presents a variant of a model trained to produce STFT coefficients. They recognized the significance of adversarial objective in preventing robotic sound quality, however they were unable to train it successfully due to its inherent instability. On the other hand, iSTFTNet (Kaneko et al., 2022) proposes modifications to HiFi-GAN, enabling it to return magnitude and phase spectrum. However, their optimal model only replaces the last two upsample blocks with inverse STFT, leaving the majority of the upsampling to be realized with transposed convolutions. They find that replacing more upsampling layers drastically degrades the quality. Pasini & Schlüter (2022) were able to successfully model the magnitude and phase spectrum of audio with higher frequency resolution, although it required multi-step training (Caillon & Esling, 2021), because of the adversarial objective instability. Also, the initial studies using GANs to generate invertible spectrograms involved estimating instantaneous frequency (Engel et al., 2018). However, these were limited to a single dataset containing only individual musical instrument notes, with the assumption of a constant instantaneous frequency.

## 3 VOCOS

### 3.1 OVERVIEW

At its core, the proposed GAN model uses Fourier-based time-frequency representation as the target data distribution for the generator. Vocos is constructed without any transposed convolutions; instead,

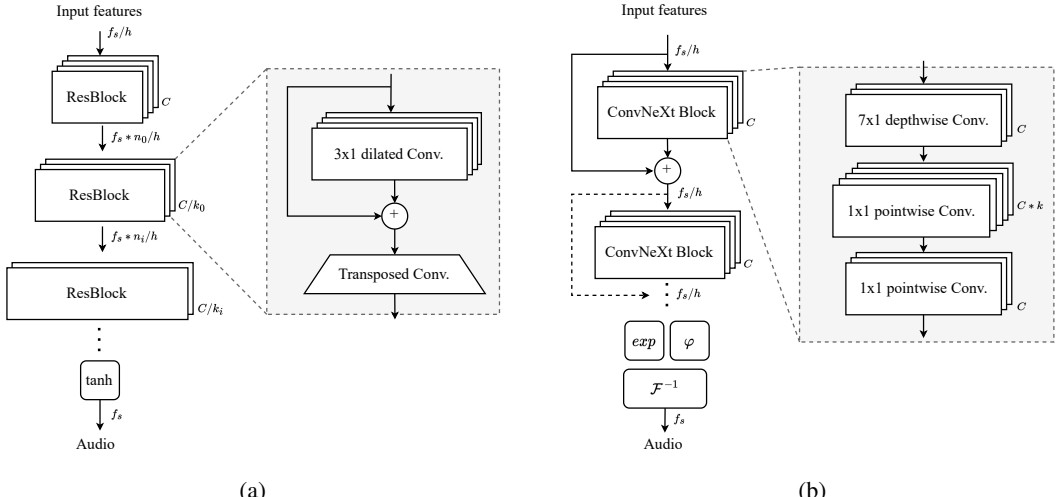

Figure 2: Comparison of a typical time-domain GAN vocoder (a), with the proposed Vocos architecture (b) that maintains the same temporal resolution across all layers. Time-domain vocoders use transposed convolutions to sequentially upsample the signal to the desired sample rate. In contrast, Vocos achieves this by using a computationally efficient inverse Fourier transform.

the upsample operation is realized solely through the fast inverse STFT. This approach permits a unique model design compared to time-domain vocoders, which typically employ a series of upsampling layers to inflate input features to the target waveform's resolution, often necessitating upscaling by several hundred times. In contrast, Vocos maintains the same temporal resolution throughout the network (Figure 2). This design, known as an isotropic architecture, has been found to work well in various settings, including Transformer (Vaswani et al., 2017). This approach can also be particularly beneficial for audio synthesis. Traditional methods often use transposed convolutions that can introduce aliasing artifacts, necessitating additional measures to mitigate the issue (Karras et al., 2021; Lee et al., 2022). Vocos eliminates learnable upsampling layers, and instead employs the well-establish inverse Fourier transform to reconstruct the original-scale waveform. In the context of converting mel-spectrograms into audio signal, the temporal resolution is dictated by the hop size of the STFT.

Vocos uses the Short-Time Fourier Transform (STFT) to represent audio signals in the time-frequency domain:

$$\text{STFT}_x[m, k] = \sum_{n=0}^{N-1} x[n]w[n-m]e^{-j2\pi kn/N} \tag{1}$$

The STFT applies the Fourier transform to successive windowed sections of the signal. In practice, the STFT is computed by taking a sequence of Fast Fourier Transforms (FFTs) on overlapping, windowed frames of data, which are created as the window function advances or "hops" through time.

## 3.2 MODEL

**Backbone**  Vocos adapts ConvNeXt (Liu et al., 2022) as the foundational backbone for the generator. It first embeds the input features into a hidden dimensionality and then applies a stack of 1D convolutional blocks. Each block consists of a depthwise convolution, followed by an inverted bottleneck that projects features into a higher dimensionality using pointwise convolution. GELU (Gaussian Error Linear Unit) activations are used within the bottleneck, and Layer Normalization is employed between the blocks.

**Head**    Fourier transform of real-valued signals is conjugate symmetric, so we use only a single side band spectrum, resulting in $n_{fft}/2 + 1$ coefficients per frame. As we parameterize the model to output phase and magnitude values, hidden-dim activations are projected into a tensor $\mathbf{h}$ with $n_{fft} + 2$ channels and splitted into:

$$\mathbf{m}, \mathbf{p} = \mathbf{h}[1 : (n_{fft}/2 + 1)], \mathbf{h}[(n_{fft}/2 + 2) : n]$$

To represent the magnitude, we apply the exponential function to $\mathbf{m}$: $\mathbf{M} = \exp(\mathbf{m})$.

We map $\mathbf{p}$ onto the unit circle by calculating the cosine and sine of $\mathbf{p}$ to obtain $\mathbf{x}$ and $\mathbf{y}$, respectively:

$$\mathbf{x} = \cos(\mathbf{p})$$
$$\mathbf{y} = \sin(\mathbf{p})$$

Finally, we represent complex-valued coefficients as: $\text{STFT} = \mathbf{M} \cdot (\mathbf{x} + j\mathbf{y})$.

Importantly, this simple formulation allows to express phase angle $\varphi = \text{atan2}(\mathbf{y}, \mathbf{x})$ for any real argument $\mathbf{p}$, and it ensures that $\varphi$ is correctly wrapped into the desired range $(-\pi, \pi]$.

**Discriminator**    We employ the multi-period discriminator (MPD) as defined by Kong et al. (2020), and multi-resolution discriminator (MRD) (Jang et al., 2021).

## 3.3 Loss

Following the approach proposed by Kong et al. (2020), the training objective of Vocos consists of reconstruction loss, adversarial loss and feature matching loss. However, we adopt a hinge loss formulation instead of the least squares GAN objective, as suggested by Zeghidour et al. (2021):

$$\ell_G(\hat{\boldsymbol{x}}) = \frac{1}{K} \sum_k \max\left(0, 1 - D_k(\hat{\boldsymbol{x}})\right)$$

$$\ell_D(\boldsymbol{x}, \hat{\boldsymbol{x}}) = \frac{1}{K} \sum_k \max\left(0, 1 - D_k(\boldsymbol{x})\right) + \max\left(0, 1 + D_k(\hat{\boldsymbol{x}})\right)$$

where $D_k$ is the $k$th subdiscriminator. The reconstruction loss, denoted as $L_{mel}$, is defined as the L1 distance between the mel-scaled magnitude spectrograms of the ground truth sample $\boldsymbol{x}$ and the synthesized sample: $\hat{\boldsymbol{x}}$: $L_{mel} = \|\mathcal{M}(\boldsymbol{x}) - \mathcal{M}(\hat{\boldsymbol{x}})\|_1$. The feature matching loss, denoted as $L_{feat}$ is calculated as the mean of the distances between the $l$th feature maps of the $k$th subdistriminator: $L_{feat} = \frac{1}{KL} \sum_k \sum_l \left\|D_k^l(\boldsymbol{x}) - D_k^l(\hat{\boldsymbol{x}})\right\|_1$.

# 4 Results

## 4.1 Mel-spectrograms

Reconstructing audio waveforms from mel-spectrograms has become a fundamental task for vocoders in contemporary speech synthesis pipelines. In this section, we assess the performance of Vocos relative to established baseline methods.

**Data**    The models are trained on the LibriTTS dataset (Zen et al., 2019), from which we use the entire training subset (both `train-clean` and `train-other`). We maintain the original sampling rate of 24 kHz for the audio files. For each audio sample, we compute mel-scaled spectrograms using parameters: $n_{fft} = 1024$, $hop_n = 256$, and the number of Mel bins is set to 100. A random gain is applied to the audio samples, resulting in a maximum level between -1 and -6 dBFS.

**Training Details**    We train our models up to 2 million iterations, with 1 million iterations per generator and discriminator. During training, we randomly crop the audio samples to 16384 samples and use a batch size of 16. The model is optimized using the AdamW optimizer with an initial learning rate of 2e-4 and betas set to (0.9, 0.999). The learning rate is decayed following a cosine schedule.

Table 1: Objective evaluation metrics for various models, including baseline models (HiFi-GAN, iSTFTNet, BigVGAN) and Vocos.

| | UTMOS ($\uparrow$) | VISQOL ($\uparrow$) | PESQ ($\uparrow$) | V/UV F1 ($\uparrow$) | Periodicity ($\downarrow$) |
|---|---|---|---|---|---|
| Ground truth | 4.058 | – | – | – | – |
| HiFi-GAN | 3.669 | 4.57 | 3.093 | 0.9457 | 0.129 |
| iSTFTNet | 3.564 | 4.56 | 2.942 | 0.9372 | 0.141 |
| BigVGAN | **3.749** | 4.65 | 3.693 | 0.9557 | 0.108 |
| Vocos | 3.734 | **4.66** | **3.70** | **0.9582** | **0.101** |
|    w/ absolute phase | 3.590 | 4.65 | 3.565 | 0.9556 | 0.108 |
|    w/ Snake | 3.699 | 4.66 | 3.629 | 0.9579 | 0.102 |
|    w/o ConvNeXt | 3.658 | 4.65 | 3.528 | 0.9534 | 0.109 |

**Baseline Methods**  Our proposed model, Vocos, is compared to: iSTFTNet (Kaneko et al., 2022), BigVGAN (Lee et al., 2022), and HiFi-GAN (Kong et al., 2020). These models are retrained on the same LibriTTS subset for up to 2 million iterations, following the original training details recommended by the authors. We use the official implementations of BigVGAN[1] and HiFi-GAN[2], and a community open-sourced version of iSTFTNet[3].

### 4.1.1 EVALUATION

**Objective Evaluation**  For objective evaluation of our models, we employ the UTMOS (Saeki et al., 2022) automatic Mean Opinion Score (MOS) prediction system. Although UTMOS can yield scores highly correlated with human evaluations, it is restricted to 16 kHz sample rate. To assess perceptual quality, we also utilize ViSQOL (Chinen et al., 2020) in audio-mode, which operates in the full band. Our evaluation process also encompasses several other metrics, including the Perceptual Evaluation of Speech Quality (PESQ) (Rix et al., 2001), periodicity error, and the F1 score for voiced/unvoiced classification (V/UV F1), following the methodology proposed by Morrison et al. (2021). The results are presented in Table 1. Vocos achieves superior performance in most of the metrics compared to the other models. It obtains the highest scores in VISQOL and PESQ. Importantly, it also effectively mitigates the periodicity issues frequently associated with time-domain GANs. BigVGAN stands out as the closest competitor, especially in the UTMOS metric, where it slightly outperforms Vocos.

In our ablation study, we examined the impact of specific design decisions on Vocos's performance:

- **Vocos with absolute phase**: In this variant, we predict phase angles using a $\tanh$ nonlinearity, scaled to fit within the range of $[-\pi, \pi]$. This formulation does not give the model an inductive bias regarding the periodic nature of phase, and the results show it leads to degraded quality. This finding emphasizes the importance of implicit phase wrapping in the effectiveness of Vocos.

- **Vocos with Snake activation**: Although Snake (Ziyin et al., 2020) has been shown to enhance time-domain vocoders such as BigVGAN, in our case, it did not result in performance gains; in fact, it showed a slight decline. The primary purpose of the Snake function is to induce periodicity, addressing the limitations of time-domain vocoders. Vocos, on the other hand, explicitly incorporates periodicity through the use of Fourier basis functions, eliminating the need for specialized modules like Snake.

- **Vocos without ConvNeXt**: Replacing ConvNeXt blocks with traditional ResBlocks with dilated convolutions, slightly lowers scores across all evaluated metrics. This finding highlights the integral role of ConvNeXt blocks in Vocos, contributing significantly to its overall success.

---

[1]https://github.com/NVIDIA/BigVGAN
[2]https://github.com/jik876/hifi-gan
[3]https://github.com/rishikksh20/iSTFTNet-pytorch

Table 2: Subjective evaluation metrics – 5-scale Mean Opinion Score (MOS) and Similarity Mean Opinion Score (SMOS) with 95% confidence interval.

|  | **MOS** (↑) | **SMOS** (↑) |
|---|---|---|
| Ground truth | 3.81±0.16 | 4.70±0.11 |
| HiFi-GAN | 3.54±0.16 | 4.49±0.14 |
| iSTFTNet | 3.57±0.16 | 4.42±0.16 |
| BigVGAN | 3.64±0.15 | 4.54±0.14 |
| Vocos | 3.62±0.15 | 4.55±0.15 |

**Subjective Evaluation** We conducted crowd-sourced subjective assessments, using a 5-point Mean Opinion Score (MOS) to evaluate the naturalness of the presented recordings. Participants rated speech samples on a scale from 1 ('poor - completely unnatural speech') to 5 ('excellent - completely natural speech'). Following (Lee et al., 2022), we also conducted a 5-point Similarity Mean Opinion Score (SMOS) between the reproduced and ground-truth recordings. Participants were asked to assign a similarity score to pairs of audio files, with a rating of 5 indicating 'Extremely similar' and a rating of 1 representing 'Not at all similar'.

To ensure the quality of responses, we carefully selected participants through a third-party crowd-sourcing platform. Our criteria included the use of headphones, fluent English proficiency, and a declared interest in music listening as a hobby. A total of 1560 ratings were collected from 39 participants.

The results are detailed in Table 2. Vocos performs on par with the state-of-the-art in both perceived quality and similarity. Statistical tests show no significant differences between Vocos and BigVGAN in MOS and SMOS scores, with p-values greater than 0.05 from the Wilcoxon signed-rank test.

Table 3: VISQOL scores of various models tested on the MUSDB18 dataset. A higher VISQOL score indicates better perceptual audio quality.

|  | Mixture | Drums | Bass | Other | Vocals | Average |
|---|---|---|---|---|---|---|
| HiFi-GAN | 4.46 | 4.40 | 4.12 | 4.44 | 4.54 | 4.39 |
| iSTFTNet | 4.47 | 4.48 | 3.80 | 4.40 | 4.53 | 4.34 |
| BigVGAN | 4.60 | 4.60 | 4.29 | **4.58** | 4.64 | 4.54 |
| Vocos | **4.61** | **4.61** | **4.31** | **4.58** | **4.66** | **4.55** |

**Out-of-distribution data** A crucial aspect of a vocoder is its ability to generalize to unseen acoustic conditions. In this context, we evaluate the performance of Vocos with out-of-distribution audio using the MUSDB18 dataset (Rafii et al., 2017), which includes a variety of multi-track music audio like vocals, drums, bass, and other instruments, along with the original mixture. The VISQOL scores for this evaluation are provided in Table 3. From the table, Vocos consistently outperforms the other models, achieving the highest scores across all categories.

Figure 3 presents spectrogram visualization of an out-of-distribution singing voice sample, as reproduced by different models. Periodicity artifacts are commonly observed when employing time-domain GANs. BigVGAN, with its anti-aliasing filters, is able to recover some of the harmonics in the upper frequency ranges, marking an improvement over HiFi-GAN. Nonetheless, Vocos appears to provide a more accurate reconstruction of these harmonics, without the need for additional modules.

## 4.2 NEURAL AUDIO CODEC

While traditionally, neural vocoders reconstruct the audio waveform from a mel-scaled spectrogram – an approach widely adopted in many speech synthesis pipelines – recent research has started to utilize learnt features (Siuzdak et al., 2022), often in a quantized form (Borsos et al., 2022).

In this section, we draw a comparison with EnCodec (Défossez et al., 2022), an open-source neural audio codec, which follows a typical time-domain GAN vocoder architecture and uses Residual Vector Quantization (RVQ) (Zeghidour et al., 2021) to compress the latent space. RVQ cascades multiple layers of Vector Quantization, iteratively quantizing the residuals from the previous stage to form a multi-stage structure, thereby enabling support for multiple bandwidth targets. In EnCodec, dedicated discriminators are trained for each bandwidth. In contrast, we have adapted Vocos to be a conditional GAN with a projection discriminator (Miyato & Koyama, 2018), and have incorporated adaptive layer normalization (Huang & Belongie, 2017) into the generator.

**Audio reconstruction**   We utilize the open-source model checkpoint of EnCodec operating at 24 kHz. To align with EnCodec, we scale down Vocos to match its parameter count (7.9M) and train it on clean speech segments sourced from the DNS Challenge (Dubey et al., 2022). Our evaluation, conducted on the DAPS dataset (Mysore, 2014) and detailed in Table 4, reveals that despite EnCodec's reconstruction artifacts not significantly impacting PESQ and Periodicity scores, they are considerably reflected in the perceptual score, as denoted by UTMOS. In this regard, Vocos notably outperforms EnCodec. We also performed a crowd-sourced subjective assessment to evaluate the naturalness of these samples. The results, as shown in Table 5, indicate that Vocos consistently achieves better performance across a range of bandwidths, based on evaluations by human listeners.

Table 4: Objective evaluation metric calculated for various bandwidths.

|  | Bandwidth | UTMOS ($\uparrow$) | VISQOL ($\uparrow$) | PESQ ($\uparrow$) | V/UV F1 ($\uparrow$) | Periodicity ($\downarrow$) |
|---|---|---|---|---|---|---|
| EnCodec | 1.5 kbps | 1.527 | 3.74 | 1.508 | 0.8826 | 0.215 |
|  | 3.0 kbps | 2.522 | 3.93 | 2.006 | 0.9347 | 0.141 |
|  | 6.0 kbps | 3.262 | 4.13 | 2.665 | 0.9625 | 0.090 |
|  | 12.0 kbps | 3.765 | 4.25 | **3.283** | **0.9766** | **0.062** |
| Vocos | 1.5 kbps | 3.210 | 3.88 | 1.845 | 0.9238 | 0.160 |
|  | 3.0 kbps | 3.688 | 4.06 | 2.317 | 0.9380 | 0.135 |
|  | 6.0 kbps | 3.822 | 4.22 | 2.650 | 0.9439 | 0.124 |
|  | 12.0 kbps | **3.882** | **4.34** | 2.874 | 0.9482 | 0.116 |

Table 5: Subjective evaluation metrics – 5-scale Mean Opinion Score (MOS) with 95% confidence interval for various bandwidths.

| Bandwidth | Vocos | EnCodec |
|---|---|---|
| 1.5 kbps | **2.73**±0.20 | 1.09±0.05 |
| 3 kbps | **3.50**±0.18 | 1.71±0.21 |
| 6 kbps | **3.84**±0.16 | 2.41±0.15 |
| 12 kbps | **4.00**±0.16 | 3.08±0.19 |
| **Ground truth** | 4.09±0.16 | |

**End-to-end text-to-speech**   Recent progress in text-to-speech (TTS) has been notably driven by language modeling architectures employing discrete audio tokens. Bark (Suno AI, 2023), a widely recognized open-source model, leverages a GPT-style, decoder-only architecture, with EnCodec's 6kbps audio tokens serving as its vocabulary. Vocos trained to reconstruct EnCodec tokens can effectively serve as a drop-in replacement vocoder for Bark. We have provided text-to-speech samples from Bark and Vocos on our website and encourage readers to listen to them for a direct comparison.[4].

## 4.3   INFERENCE SPEED

Our inference speed benchmarks were conducted using an Nvidia Tesla A100 GPU and an AMD EPYC 7542 CPU. The code was implemented in Pytorch, with no hardware-specific optimizations.

---

[4]Listen to audio samples at `https://gemelo-ai.github.io/vocos/`

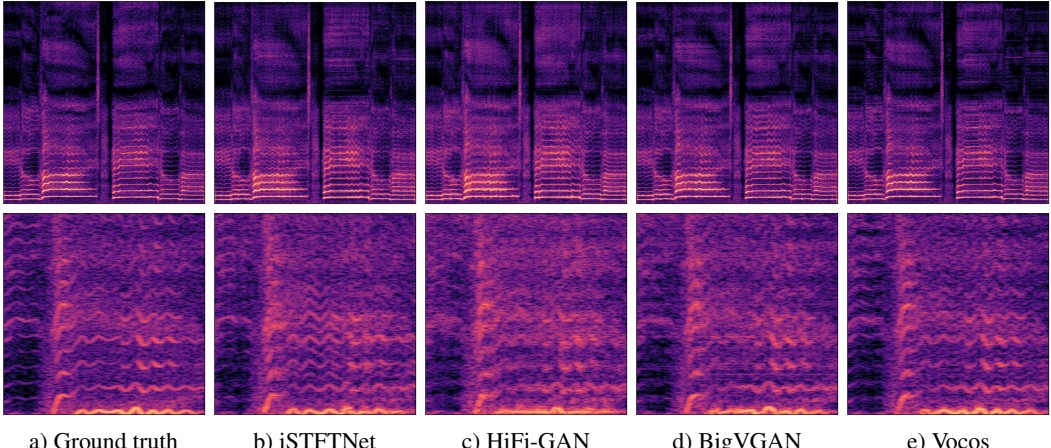

| a) Ground truth | b) iSTFTNet | c) HiFi-GAN | d) BigVGAN | e) Vocos |

Figure 3: Spectrogram visualization of an out-of-distribution singing voice sample reproduced by different models. The bottom row presents a zoomed-in view of the upper midrange frequency range.

The forward pass was computed using a batch of 16 samples, each one second long. Table 6 presents the synthesis speed and model footprint of Vocos in comparison to other models.

Vocos showcases notable speed advantages compared to other models, operating approximately 13 times faster than HiFi-GAN and nearly 70 times faster than BigVGAN. This speed advantage is particularly pronounced when running without GPU acceleration. This is primarily due to the use of the Inverse Short-Time Fourier Transform (ISTFT) algorithm instead of transposed convolutions. We also evaluate a variant of Vocos that utilizes ResBlock's dilated convolutions instead of ConvNeXt blocks. Depthwise separable convolutions offer an additional speedup when executed on a GPU.

Table 6: Model footprint and synthesis speed. xRT denotes the speed factor relative to real-time. A higher xRT value means the model can generate speech faster than real-time, with a value of 1.0 denoting real-time speed.

| Model | xRT (↑) | | Parameters |
|---|---|---|---|
| | GPU | CPU | |
| HiFi-GAN | 495.54 | 5.84 | 14.0 M |
| BigVGAN | 98.61 | 0.40 | 14.0 M |
| ISTFTNet | 1045.94 | 14.44 | 13.3 M |
| Vocos | **6696.52** | 169.63 | 13.5 M |
| w/o ConvNeXt | 4565.71 | **193.56** | 14.9 M |

## 5  CONCLUSIONS

This paper introduces Vocos, a novel neural vocoder that bridges the gap between time-domain and Fourier-based approaches. Vocos tackles the challenges associated with direct reconstruction of complex-valued spectrograms, with careful design of generator that correctly handle phase wrapping. It achieves accurate reconstruction of the coefficients in Fourier-based time-frequency representations.

The results demonstrate that the proposed vocoder matches state-of-the-art audio quality while effectively mitigating periodicity issues commonly observed in time-domain GANs. Importantly, Vocos provides a significant computational efficiency advantage over traditional time-domain methods by utilizing inverse fast Fourier transform for upsampling.

Overall, the findings of this study contribute to the advancement of neural vocoding techniques by incorporating the benefits of Fourier-based time-frequency representations. The open-sourcing of the source code and model weights allows for further exploration and application of the proposed vocoder in various audio processing tasks.

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

## A   MODIFIED DISCRETE COSINE TRANSFORM (MDCT)

While STFT is widely used in audio processing, there are other time-frequency representations with different properties. In audio coding applications, it is desirable to design the analysis/synthesis system in such a way that the overall rate at the output of the analysis stage equals the rate of the input signal. Such systems are described as being critically sampled. When we transform the signal via the DFT, even a slight overlap between adjacent blocks increases the data rate of the spectral representation of the signal. With 50% overlap between adjoining blocks, we end up doubling our data rate.

The Modified Discrete Cosine Transform (MDCT) with its corresponding Inverse Transform (IMDCT) have become a crucial tool in high-quality audio coding as they enable the implementation of a critically sampled analysis/synthesis filter bank. A key feature of these transforms is the Time-Domain Aliasing Cancellation (TDAC) property, which allows for the perfect reconstruction of overlapping segments from a source signal.

The MDCT is defined as follows:

$$X[k] = \sum_{n=0}^{2N-1} x[n] \cos \left[ \frac{\pi}{N} \left( n + \frac{1}{2} + \frac{N}{2} \right) \left( k + \frac{1}{2} \right) \right] \tag{2}$$

for $k = 0, 1, \ldots, N-1$ and $N$ is the length of the window.

The MDCT is a lapped transform and thus produces $N$ output coefficients from $2N$ input samples, allowing for a 50% overlap between blocks without increasing the data rate.

There is a relationship between the MDCT and the DFT through the Shifted Discrete Fourier Transform (SDFT) (Wang & Vilermo, 2003). It can be leveraged to implement a fast version of the MDCT using FFT (Bosi & Goldberg, 2002). See Appendix A.3.

### A.1   VOCOS AND MDCT

MDCT is attractive in audio coding because of its its efficiency and compact representation of audio signals. In the context of deep learning, this might be seen as reduced dimensionality, potentially advantageous as it requires fewer data points during generation.

While STFT coefficients can be conveniently expressed in polar form, providing a clear interpretation of both magnitude and phase, MDCT represents the signal only in a real subspace of the complex space needed to accurately convey spectral magnitude and phase. Naive approach would be to treat raw unnormalized hidden outputs of the network as MDCT coefficients and convert it back to time-domain with IMDCT. In our preliminary experiments we found that it led to slower convergence. However we can easily observe that the MDCT spectrum, similarly to the STFT, can be more perceptually meaningful on the logarithmic scale, which reflects the logarithmic nature of human auditory perception of sound intensity. But as the MDCT can take also negative values, they cannot be represented using the conventional logarithmic transformation.

One solution is to utilize a symmetric logarithmic function. In the context of deep learning, Hafner et al. (2023) introduces such function and its inverse, referred to as symlog and symexp respectively:

$$\text{symlog}(x) = \text{sign}(x) \ln(|x| + 1) \qquad \text{symexp}(x) = \text{sign}(x)(\exp(|x|) - 1) \tag{3}$$

The symlog function compresses the magnitudes of large values, irrespective of their sign. Unlike the conventional logarithm, it is symmetric around the origin and retains the input sign. We note the correspondence with the $\mu$-law companding algorithm, a well-established method in telecommunication and signal processing.

An alternative approach involves parametrizing the model to output the absolute value of the MDCT coefficients and its corresponding sign. While the MDCT does not directly convey information about phase relationships, this strategy may offer advantages as the sign of the MDCT can potentially provide additional insights indirectly. For example, an opposite sign could imply a phase difference

Table 7: Objective evaluation metrics for MDCT variant of Vocos compared to the ISTFT baseline.

| | UTMOS ($\uparrow$) | PESQ ($\uparrow$) | V/UV F1 ($\uparrow$) | Periodicity ($\downarrow$) |
|---|---|---|---|---|
| Ground truth | 4.058 | – | – | – |
| Baseline (ISTFT) | **3.734** | **3.70** | **0.9582** | **0.101** |
| IMDCT (symexp) | 3.498 | 3.648 | 0.9569 | 0.106 |
| IMDCT (sign) | 3.536 | 3.565 | 0.9547 | 0.109 |

of 180 degrees. In practice, we compute a "soft" sign using the cosine activation function, which supposedly provides a periodic inductive bias. Hence, similar to the ISTFT head, this approach projects the hidden activations into two values for each frequency bin, representing the final coefficients as $\text{MDCT} = \exp(\mathbf{m}) \cdot \cos(\mathbf{p})$.

## A.2 RESULTS

Table 7 presents objective evaluation metrics for a variant of Vocos that represents audio samples with MDCT coefficients. Both 'symexp' and 'sign' demonstrate significantly weaker performance compared to their STFT-based counterpart. This suggests that while MDCT may be attractive in audio coding applications, its properties may not be as favorable in the context of generative modeling with GANs. The redundancy inherent in the STFT representation appears to be beneficial for generative tasks. This finding aligns with the work of Gritsenko et al. (2020), who discovered that an overcomplete Fourier basis contributed to improved training stability. Furthermore, it is worth noting that the MDCT, being a lapped transform, incorporates information from surrounding windows, which effectively act as aliases of the signal. To ensure Time Domain Alias Cancellation (TDAC), the prediction of the coefficients has to be accurate and consistent over the frames.

## A.3 FORWARD MDCT ALGORITHM

---
**Algorithm 1** Fast MDCT Algorithm realized with FFT

---
1: **Input:** Audio signal $x$ with frame length $N$
2: **Output:** MDCT coefficients $X$
3: **procedure** MDCT($x$)
4:     **for** each frame $f$ in $x$ with overlap of $N/2$ **do**
5:         $f \leftarrow f \times$ window function
6:         $f \leftarrow f \times e^{-j\frac{2\pi n}{2N}}$                            $\triangleright$ Pre-twiddle
7:         $f \leftarrow \text{FFT}(f)$                           $\triangleright$ N-point FFT
8:         $f \leftarrow f \times e^{-j\frac{2\pi}{N}n_0\left(k+\frac{1}{2}\right)}$           $\triangleright$ Post-twiddle
9:         $f \leftarrow f \times \sqrt{\frac{1}{N}}$
10:        $X_k \leftarrow \Re(f) \times \sqrt{2}$
11:     **end for**
12:     **return** $X$
13: **end procedure**

---

