# OpenReview forum: "Vocos: Closing the gap between time-domain and Fourier-based neural vocoders for high-quality audio synthesis"
_ICLR.cc/2024/Conference — ICLR 2024 poster_

### Official Review · Reviewer_9BcD · 2023-10-26

**Soundness:** 3 good
**Presentation:** 4 excellent
**Contribution:** 3 good
**Rating:** 6
**Confidence:** 3

**Summary:**

The paper presents a new method for neural vocoding by generateing fourier spectral coefficients instead of generating wave forms in the time-domain, this leads to significant computational speed up.

**Strengths:**

- The paper is well-written.
- The authors release the source code.
- The authors present an extensive evaluation of the Vocoder.
- The proposed models offer significant speed-ups compared to the related work without loss in the quality.
- The proposed method is very simple and leads to impressive results.

**Weaknesses:**

1- Limited ablation study: The authors show the effect of replacing ConvNeXt with residual blocks, but no studies are presented on the architecture itself: the effect of the number of layers and hidden dimensions.  The effects of the resolution and overlap of the mel spectrograms.

2- Limited explanation of the backbone architecture: I think the way the backbone network is processing the mel-spectrograms is still ambigious. After digging into the code, it seems that the frequency dimension of the spectrogram is used as input channels, and 1D convolution is performed over the time dimension. The receptive field of each timeframe grows by 7 frames in each layer (right?). Therefore, the decoding of one timeframe depends on the adjacent time frames. This is controlled by the receptive field of the last layer. This raises a lot of questions: (a) Why did you choose 7? (b) how increasing the depth of the backbone network (and therefore the receptive field of the last layer over time) affects the performance.

3- The encodec experiments Table 4 is not clear for me: How did you control the bandwidth for your model? Is it by reducing the mel-spectrogram resolution?

4- In end-to-end text-to-speech: You write, " Vocos trained to reconstruct EnCodec tokens". This is not clear: are you using Vocos) to decode EnCodec tokens?

5- (minor) Table 5 Inference speed: Is it possible to include float or multiply-accumulate operations since there is a chance that some FFT operations can be driver (or hardware) accelerated?

**Questions:**

Can you please address the weaknesses section.

---

> ### Author Response · Authors · 2023-11-23
>
> Thank you for your insightful feedback. We appreciate the time you took to delve into the code to investigate the details.
>
> > the way the backbone network is processing the mel-spectrograms is still ambigious. After digging into the code, it seems that the frequency dimension of the spectrogram is used as input channels, and 1D convolution is performed over the time dimension.
>
> You are correct, and we realize that our paper did not clearly state that we employ 1D convolution layers, as mel-spectrograms are often processed as images. We have now made this clearer in the revised manuscript.
>
> > Therefore, the decoding of one timeframe depends on the adjacent time frames. This is controlled by the receptive field of the last layer. This raises a lot of questions: (a) Why did you choose 7? (b) how increasing the depth of the backbone network (and therefore the receptive field of the last layer over time) affects the performance.
>
> These are interesting questions. You are correct about the receptive field expanding with each layer. With a kernel size of 7 and the number of layers in our proposed config, the receptive field extends to approximately half a second. This aligns with other works in the time-domain, such as HiFi-GAN. We argue that increasing the depth, and consequently the receptive field, might not necessarily enhance performance, as vocoders typically model local auditory patterns. However, this should be methodically evaluated, and we believe it might be explored as future work. In this study we focused on comparing Vocos to other popular neural vocoders within the same parameters budget.
>
> Although we do not directly examine the performance of scaled-down versions of Vocos, please note that in the Section 4.2 we present results from a 7.9M parameters model (to align with EnCodec). This model has approximately 40% fewer parameters compared to the baseline 13.5M model.
>
> > The encodec experiments Table 4 is not clear for me: How did you control the bandwidth for your model?
>
> EnCodec functions as an autoencoder with a hierarchical vector-quantized bottleneck. It can both encode audio into a quantized representation and reconstruct (decode) a waveform from it. The bandwidth is controlled by varying the number of codes preserved from the codebook. We trained Vocos to serve as a decoder of these tokens. Codes of different bandwidths essentially mean different acoustic features, kind of like how mel-spectrograms can have 80, 100, or more mel bins.

---

### Official Review · Reviewer_cFuH · 2023-10-30

**Soundness:** 3 good
**Presentation:** 3 good
**Contribution:** 2 fair
**Rating:** 5
**Confidence:** 5

**Summary:**

This paper presents an approach to Neural Vocoding with GAN based on generating directly Fourier coefficients instead of targeting time domain signals. Overall, the paper is very well written and provides a very large set of (both qualitative and quantitative) experiments and evaluations, demonstrating the soundness of the proposed approach.

Although I think that this work is of very high quality, and I agree overall with the argument made by the authors in the spectral/temporal dilemma in generative models (which has been a central question of study in the audio domain), I am not convinced by the scientific novelty of the proposed method. For instance, the GANSynth model proposed in 2019 (but not cited in this paper) already proposed a GAN model to produce invertible spectral representations, by modeling instantaneous frequencies:
Engel, J., Agrawal, K. K., Chen, S., Gulrajani, I., Donahue, C., & Roberts, A. (2019). Gansynth: Adversarial neural audio synthesis. arXiv preprint arXiv:1902.08710.
https://openreview.net/pdf?id=H1xQVn09FX

Hence, the major novelty proposed by the paper is to use « ConvNeXt blocks », which have themselves been introduced in a previous paper (Liu et al. 2022). Therefore, it seems rather a slim contribution to be accepted in the ICLR conference. I would advise the author to resubmit the paper in a more audio-applicative type of conference.

**Strengths:**

- Very large array of experiments demonstrating the quality of the approach.
- Sound reasoning and scientific method for establishing the overall architecture.
- Targeting efficient and lightweight models is always a welcome addition in the current trends of deep generative modelling research.

**Weaknesses:**

- Low amount of scientific novelty
- Although I agree that GANs are still dominant, neural vocoders are increasingly being driven by diffusion models over the past months.
- Pictures of spectrograms are usually unhelpful in paper and in this case, I feel it is even harder to truly understand what is the added value of this figure.

**Questions:**

-

---

> ### Author Response · Authors · 2023-11-23
>
> Thank you for your review and for highlighting the relevance of GANSynth to our work, which we regrettably overlooked in our initial submission. We have revised the manuscript accordingly to include this reference. Nevertheless, there are important distinctions between GANSynth and our proposed Vocos model, and we'd like to clarify these in response to your concerns:
>
> While GANSynth indeed addresses the generation of invertible spectral representations, its focus is limited to the NSynth dataset, which consists solely of individual musical notes. This allows the authors to assume a constant instantaneous frequency. However, this is not true for real-life audio signals (which may have time-varying frequencies), like speech. GANSynth cannot be trivially extended to a broader class of audio signals, and this was left as a future work (see authors' conclusion). But actually it never happened, and shortly after, time-domain GANs for audio took off. MelGAN and subsequent works have demonstrated that, with advanced critics, it is possible to directly model coherent audio in the time domain, alleviating the need for challenging phase estimation. This approach has become the de facto standard in neural vocoding.
>
> Vocos revisits phase (and magnitude) estimation with a straightforward formulation that enables implicit phase wrapping. To clarify and further support our proposed method, we present a new ablation study in which we estimate (scaled) phase angles using the tanh nonlinearity. Please refer to the updated paper, where we demonstrate that this approach results in reduced audio quality (see Table 1). This formulation does not give the model an implicit bias regarding the periodic nature of phase. Vocos, on the other hand, proposes to define activation function in terms of a unit circle, which produces meaningful values for any phase angles. We experimentally demonstrate that this simple formulation allows for successful phase estimation in a non-autoregressive setup.
>
> We disagree with the reviewer's view that the major novelty of our paper is the use of ConvNeXt blocks. Vocos, even without ConvNeXt, demonstrates comparable performance to HiFi-GAN, while being an order of magnitude faster. We've made updates to the paper to better clarify the contributions of this work (see Section 1.2).

---

### Official Review · Reviewer_7A8e · 2023-11-01

**Soundness:** 4 excellent
**Presentation:** 3 good
**Contribution:** 4 excellent
**Rating:** 8
**Confidence:** 5

**Summary:**

Vocos is a waveform synthesis (vocoder) model operating directly on the frequency scale to estimate Fourier spectral coefficients (magnitude and phase) followed by inverse Fourier transform (iSTFT). Due to the challenges in phase modeling with neural networks, previous non-autoregressive, GAN-based neural vocoders directly generate time-domain waveforms through an upsampling architecture, leading to a slowdown in inference latency (even with the non-autoregressive modeling). Vocos, to the best of the reviewer's knowledge, constitutes the first success in the direct estimation of both the magnitude and phase entirely from the frequency domain, resulting in a highly efficient neural vocoder.

**Strengths:**

As stated in the summary, direct phase estimation has historically been known to be challenging. The major contribution of Vocos is being the first model to succeed in high-quality phase estimation for neural vocoding through the careful design of a sinusoidal activation head that handles phase wrapping. Since the feature propagation is entirely in the spectral domain with the same I/O dimension, Vocos is significantly faster than time-domain models and more flexible in adopting modern isotropic architecture (ConvNeXt), previously proposed in other data domains, for neural vocoding tasks. Keeping the scope of a general-purpose neural vocoder is also a plus, where the proposed method is not dependent on a specific input representation (mel spectrogram) and can also be trained with the latent feature as a neural audio codec.

**Weaknesses:**

While the main focus of the manuscript is realizing an efficient neural vocoder operating in the frequency domain, it would also be interesting to further assess the robustness of the proposed method. Considering that the experimental setup largely follows a previous time-domain neural vocoder (BigVGAN), readers may wonder if Vocos can also achieve similar robustness. The provided samples in the manuscript only contain clean speech samples. To further convince the readers, it would be useful to add non-clean speech and audio samples to the demo (noisy speech, environmental sounds, and music, for example) similar to the results presented in the previous work. The VISQOL score in Table 3 looks promising, so letting the readers form their opinion on the subjective quality by adding such samples will be helpful as well. By adding these samples, we could also confirm that the objective VISQOL scores align with human perception as well.

In my opinion, the statement "ConvNeXt blocks can more effectively model spatially local input patterns" is not fully supported by current results; while it is true that Vocos can easily adopt modern isotropic architectures, the same can also be explored with time-domain neural vocoders. To verify this under scrutiny, adopting ConvNeXt to the time-domain GAN vocoder and measuring the performance gain/loss will be useful (probably with dilation to the depthwise conv in ConvNeXt as done by ResBlock in HiFi-GAN). If the time-domain GAN vocoder shows degradation in performance, it will make the benefits of isotropy in Vocos more convincing.

**Questions:**

Overall, Vocos manifests an exciting step towards enabling a neural vocoder operating entirely in the frequency domain; its fast speed will contribute to accelerating efficient speech synthesis solutions, where the speed of such models has been bottlenecked by the time-domain neural vocoder. But at the same time, I am also interested in the scalability of the method. Specifically, a neural vocoder is also recently viewed as a (de)compression model (similar to VQGAN in the image domain, with EnCodec and DAC[1] as examples) to build the audio generative model in the latent space. For this application, practitioners may want higher quality decoding with increased scale if speed is not a concern. Can we expect that the Vocos (either as a standalone vocoder or the decompression model) can be scaled up further to achieve even better quality? Or, if Vocos exhibits a performance limit at a certain scale, what would be the root cause in the author's opinion?

[1] Kumar, Rithesh, et al. "High-Fidelity Audio Compression with Improved RVQGAN." arXiv preprint arXiv:2306.06546 (2023).

---

> ### Author Response · Authors · 2023-11-23
>
> Thank you for expressing support for this paper! Below we address your specific points.
>
> > The provided samples in the manuscript only contain clean speech samples. To further convince the readers, it would be useful to add non-clean speech and audio samples to the demo
>
> This is a good point, we have now uploaded noisy samples from the LibriTTS `test-other` evaluation set. You can find them here: https://anonymous5425.github.io/anonymous-submission/#libritts-test-other
>
> > In my opinion, the statement "ConvNeXt blocks can more effectively model spatially local input patterns" is not fully supported by current results; while it is true that Vocos can easily adopt modern isotropic architectures, the same can also be explored with time-domain neural vocoders.
>
> We added that sentence, as we observed extra improvements in our setup with ConvNeXt blocks. We do not claim that this will necessarily generalize to other architectures. We have rephrased this in the paper to make it clearer.
>
> > adopting ConvNeXt to the time-domain GAN vocoder and measuring the performance gain/loss will be useful (probably with dilation to the depthwise conv in ConvNeXt as done by ResBlock in HiFi-GAN). If the time-domain GAN vocoder shows degradation in performance, it will make the benefits of isotropy in Vocos more convincing.
>
> Vocos achieves isotropy by maintaining the same temporal resolution for input and output features. It cannot be easily achieved in time-domain vocoders, as the sample rate of audio signal is too high for any input features to make it practical.
>
> Although it is possible that time-domain vocoders might be improved with more capable modules, we believe this is out of scope of our work.
>
> > Can we expect that the Vocos (either as a standalone vocoder or the decompression model) can be scaled up further to achieve even better quality? Or, if Vocos exhibits a performance limit at a certain scale, what would be the root cause in the author's opinion?
>
> In our informal testing we have observed that scaling up the model does not solve all the problems and there is still a significant gap to ground truth recordings, especially in out-of-distribution conditions. We believe that the root cause might be the non-autoregressive setting with the current critics. While the feature matching loss is known to significantly contribute to overall audio quality, it does have some limitations. The activations of the discriminators' early layers contain information about the initial phase, so the model can be penalized for random (but valid) phase shift. Morrison et al. [1] investigate this in more detail.
> One solution could be to encourage a one-to-many mapping of phase angles using augmentation methods like PhaseAug [2].  Another appealing option is to potentially redesign Vocos as an autoregressive model, potentially trading off speed for higher quality.
>
> [1] Morrison, Max, et al. "Chunked Autoregressive GAN for Conditional Waveform Synthesis." International Conference on Learning Representations. 2021.
>
> [2] Lee, Junhyeok, et al. "PHASEAUG: A Differentiable Augmentation for Speech Synthesis to Simulate One-to-Many Mapping." ICASSP 2023-2023 IEEE International Conference on Acoustics, Speech and Signal Processing (ICASSP). IEEE, 2023.

---

### Official Review · Reviewer_ayuF · 2023-11-01

**Soundness:** 3 good
**Presentation:** 3 good
**Contribution:** 2 fair
**Rating:** 5
**Confidence:** 3

**Summary:**

This paper introduces a GAN-based neural vocoding method that directly reconstructs Fourier spectral coefficients rather than the prevalent time-domain generation. The experiments showcase that recovering phase information not only improves the quantitative metrics but also results in higher human ratings. In addition, its efficient architecture makes it promising to be adopted in real applications.
The paper is well-written and the motivation is clearly described. The experiments section is comprehensive.

**Strengths:**

- The paper is well-structured and easy to follow.
- The motivation of bridging the gap between time-domain and time-frequency-main is sound and the results indicate the proposed method works as expected.
- The code is open-sourced and audio samples are available.

**Weaknesses:**

- Novelty is limited considering the ICLR standards.
- It's not clear to me whether using "ConvNeXt" solely in the time-domain is already bringing all the benefits stated in the paper. The experiment of using ConvNeXt in the time-domain seems missing
- There are a few overclaims in the audio reconstruction task. PESQ/UV F1/ Periodicity numbers of EnCodec still win over Vocos by a large margin at 12.0kbps. Should we still consider "Vocos notably outperforms EnCodec?

**Questions:**

- Picking "ConvNeXt" as the backbone looked a bit random to me. Did you try other architectures and decide to use ConvNeXt in the end?
- Have you run experiments on ConvNeXt in the time domain?

---

> ### Author Response · Authors · 2023-11-23
>
> Thank you for taking the time to review our work.
>
> > The experiments showcase that recovering phase information not only improves the quantitative metrics but also results in higher human ratings.
>
> We believe there is a misunderstanding here. Recovering phase information is not unique to Vocos. Every vocoder has to recover phase from acoustic features. Vocos generates both magnitude and phase spectra directly, while prevalent GAN vocoders operate in the time domain, which implicitly incorporates phase angles (and magnitudes).
>
> Reconstruction of complex-valued spectrograms with neural networks is not a new concept. However, it has historically been quite challenging, creating a gap between time-domain and Fourier-based vocoders. Here, we propose Vocos, a neural vocoder that successfully estimates magnitude and phase spectrum in a non-autoregressive setup. Its significant contribution lies in its careful design, with the activation function defined in terms of a unit circle, which produces meaningful values for any phase angles.
>
> > It's not clear to me whether using "ConvNeXt" solely in the time-domain is already bringing all the benefits stated in the paper.
>
> The benefit of Vocos is to synthesize audio 70 times faster than state-of-the-art time-domain vocoder (BigVGAN-base), while maintaining comparable performance, as demonstrated in the subjective evaluation in the paper. This is mainly because of the Fast Fourier Transform rather than differences at a layer level. We respectfully argue that applying ConvNeXt to time-domain vocoders is not relevant here.
>
> > There are a few overclaims in the audio reconstruction task. PESQ/UV F1/ Periodicity numbers of EnCodec still win over Vocos by a large margin at 12.0kbps. Should we still consider "Vocos notably outperforms EnCodec?
>
> The reconstruction artifacts of EnCodec don’t greatly affect PESQ/UV F1/ Periodicity scores, however the subjective quality is severely impacted. To support this claim we decided to perform a crowd-sourced subjective assessment to evaluate the naturalness of these samples. The results have been included in the paper and can be found in Table 5. Vocos consistently outperforms EnCodec across all bandwidths, based on evaluations by human listeners. We also encourage you to listen to the audio samples at https://anonymous5425.github.io/anonymous-submission/

---

### Author Response · Authors · 2023-11-23

Dear Reviewers,

Thank you for your thoughtful and detailed feedback on our paper. We greatly value your expertise and have found your suggestions to be very helpful in improving our work.

In response to your comments, we have made several updates to the manuscript. For ease of review, these changes have been highlighted in blue.

---

### Meta-Review · Area_Chair_7q9Y · 2023-12-05

**Metareview:**

The paper proposes a vocoder in the spectral domain, relying on the inverse Fourier transform to produce wave samples rather than directly producing wave samples. The approach is much more efficient than directly producing wave samples, and is on par with many other approaches that work in the time domain.

There is much discussion on the technical details of the approach and the design of the experiments, but the overall methodology is sound.

The authors are encouraged to further revise the submission to address the issues raised by the reviewers.

**Justification For Why Not Higher Score:**

There is room for improvement, especially on ablation studies and arguments of design choices.

**Justification For Why Not Lower Score:**

The approach has merit and achieves what the paper claims.

---

### Decision · Program_Chairs · 2024-01-16

Accept (poster)